# Could the Lung Be a Gateway for Amphotericin B to Attack the Army of Fungi?

**DOI:** 10.3390/pharmaceutics14122707

**Published:** 2022-12-03

**Authors:** Beatriz Ferreira de Carvalho Patricio, Juliana Oliveira da Silva Lopes Pereira, Michelle Alvares Sarcinelli, Bianca Portugal Tavares de Moraes, Helvécio Vinicius Antunes Rocha, Cassiano Felippe Gonçalves-de-Albuquerque

**Affiliations:** 1Pharmacology Laboratory, Biomedical Institute, Federal University of State of Rio de Janeiro, 94 Frei Caneca Street, Rio de Janeiro 20211-010, Brazil; 2Postgraduate Program in Molecular and Cell Biology, Biomedical Institute, Federal University of State of Rio de Janeiro, 94 Frei Caneca Street, Rio de Janeiro 20211-010, Brazil; 3Laboratory of Micro and Nanotechnology, Institute of Technology of Drugs, Oswaldo Cruz Foundation, Brazil Av., 4036, Rio de Janeiro 213040-361, Brazil; 4Postgraduate Program in Biotechnology, Biology Institute, Federal Fluminense University, Rua Prof. Marcos Waldemar de Freitas Reis, Niterói 24210-201, Brazil; 5Immunopharmacology Laboratory, Biomedical Institute, Federal University of State of Rio de Janeiro, 94 Frei Caneca Street, Rio de Janeiro 20211-010, Brazil

**Keywords:** pulmonary fungal infection, amphotericin B, pulmonary drug delivery, nanoformulations

## Abstract

Fungal diseases are a significant cause of morbidity and mortality worldwide, primarily affecting immunocompromised patients. *Aspergillus*, *Pneumocystis*, and *Cryptococcus* are opportunistic fungi and may cause severe lung disease. They can develop mechanisms to evade the host immune system and colonize or cause lung disease. Current fungal infection treatments constitute a few classes of antifungal drugs with significant fungi resistance development. Amphotericin B (AmB) has a broad-spectrum antifungal effect with a low incidence of resistance. However, AmB is a highly lipophilic antifungal with low solubility and permeability and is unstable in light, heat, and oxygen. Due to the difficulty of achieving adequate concentrations of AmB in the lung by intravenous administration and seeking to minimize adverse effects, nebulized AmB has been used. The pulmonary pathway has advantages such as its rapid onset of action, low metabolic activity at the site of action, ability to avoid first-pass hepatic metabolism, lower risk of adverse effects, and thin thickness of the alveolar epithelium. This paper presented different strategies for pulmonary AmB delivery, detailing the potential of nanoformulation and hoping to foster research in the field. Our finds indicate that despite an optimistic scenario for the pulmonary formulation of AmB based on the encouraging results discussed here, there is still no product registration on the FDA nor any clinical trial undergoing ClinicalTrial.gov.

## 1. Introduction

Invasive fungal diseases are a significant cause of morbidity and mortality worldwide [1], being especially critical in immunocompromised patients such as those diagnosed with cystic fibrosis, acquired immunodeficiency disease syndrome (AIDS), and transplant recipients [2,3,4,5]. They are also causes of comorbidity in patients undergoing therapy with immunosuppressive agents and prolonged use of glucocorticoids, i.e., antimicrobial chemotherapeutic agents [2,3,4]. The most common pathogens that cause fungal lung diseases are *Aspergillus*, *Cryptococcus*, and *Pneumocystis* [6,7,8]. They are challenging to diagnose and treat, leading to increased morbidity and mortality [9]. In addition, opportunistic fungi such as *Coccidioides*, *Paracoccidioides*, *Candida*, *Sporothrix*, and *Histoplasma* can cause severe lung infections that progress to dissemination, affecting other tissues and surpassing a million cases worldwide [5,6].

Invasive pulmonary aspergillosis (IPA) affects immunocompromised patients with prolonged antibiotics, leukemia, transplanted patients, prolonged hospital stays, and AIDS [3,8]. Additionally, neutropenic patients are significantly affected, with a mortality rate reaching 80% in lung transplant patients [3,8,10]. Moreover, *Aspergillus* causes other pathogenesis, such as chronic pulmonary aspergillosis (CPA), in patients with lung diseases such as asthma, cystic fibrosis, and chronic obstructive pulmonary disease [2,3,10], leading to the acceleration of lung function decline in these patients [3,5]. 

Patients with viral respiratory tract infections are affected by pulmonary fungal coinfection, especially by *Aspergillus* [1]. These coinfections are complications of viral diseases such as influenza, acute respiratory syndrome coronavirus-2 (SARS-CoV-2), respiratory syncytial virus, and adenovirus, which leads to worsening the disease and increased mortality in those patients [1]. Influenza infections are responsible for half a million deaths annually, and 3 to 5 million people develop severe influenza infections [11]. In the H1N1 outbreak in 2009, an increasing number of cases of IPA-associated influenza was reported in the literature [12,13]. The coinfection is strongly related to admission to an intensive care unit (ICU) [14], with a high mortality rate in ICU patients [15,16,17]. The emergence of SARS-CoV-2 has brought to light a challenge for the entire health system, having a tremendous socioeconomic impact [18,19]. The coronavirus disease caused by SARS-CoV-2, COVID-19 has led to millions of deaths [1,20]. Hundreds of coinfection cases with *Aspergillus* have been described in Europe, Asia, Australia, and America [1,21,22]. The mortality rate in COVID-19 patients associated with IPA is incredibly high [23]. 

Pneumonia caused by *P. jirovecii* is common in AIDS patients, accounting for half a million deaths [6]. On the other hand, the fungus *Cryptococcus* affects patients who have undergone solid organ transplantation. This infection starts in the lung and may spread to the central nervous system, causing meningitis [6]. Treating these infections with antifungal drugs is challenging because the difficulty in reaching the minimum inhibitory concentration leads to high mortality rates [3].

In organ transplants and cancer patients, the incidence of neutropenia has been increasing, which leads to a greater chance of patients being affected by fungal infections [8]. In addition, because of the lack of defenses along the bronchiolar tree, fungal invasions are more prone to occur in the lung [24]. Thus, prophylactic therapy with antifungal agents is recommended in lung-transplanted patients [8]. Finally, massive antibiotics use, including prophylactic use to prevent bacterial coinfections in immunosuppressed patients, has increased the number of fungal infections [1,8]. In addition, pulmonary fungal colonization is persistent in patients with chronic bronchitis [25]. Thus, fungal infection is a global health and economic challenge, and further studies must be conducted on managing and treating these diseases. 

## 2. Fungal Pulmonary Diseases

Herein, we will briefly discuss the pathophysiology, evasion mechanism, and virulence of three fungi, *Aspergillus*, *Cryptococcus*, and *Pneumocystis*, because they are the most common fungi airborne transmitted and cause lung disease [26]. Chronic pulmonary aspergillosis has a global prevalence of 42 per 100,000 population [27], while allergic bronchopulmonary aspergillosis in asthma has a global burden of ~4,800,000 [6]. Pneumocystis pneumonia causes more than 400,000 cases yearly [28] and HIV-associated Cryptococcosis is about ~200,000 to ~300,000 per year, with approximately 180,000 deaths [29].

*Aspergillus* conidia (spores) are ubiquitous in the environment and are exposed to oxidative and antifungal environments, which force fungal adaptation, altering its virulence [30,31]. The spores can propagate by transfer in air, colonizing the pulmonary airways and changing the status of the infection from asymptomatic to invasive. The disease’s clinical manifestations will depend on the host’s immune response [32,33]. The *A. fumigatus* complex, followed by others such as *A. flavus*, may cause allergic bronchopulmonary aspergillosis (ABPA) or asthma with fungal sensitization (SAFS), whereas CPA affects susceptible people with lung diseases [31]. Immunosuppressed people may be more affected [30,34], including patients with cancer, autoimmune or inflammatory diseases, and with immune-metabolic abnormalities [35,36,37], and severe SARS-CoV-2 infection [38,39].

The cross-talk between epithelia and phagocytes [40] plays a critical role in the inflammatory responses to *Aspergillus* [41]. Airway epithelial cells (ECs) comprise the first line of host defense against *Aspergillus* [42,43]. Different molecules are vital to this interaction. For instance, the fucose-specific lectin of *Aspergillus fumigatus* (FleA), together with mucins, favors mucociliary clearance [44], and the galactosaminogalactan (GAG) adhesin binds to the epithelium [45]. Mutations in CFTR receptors in cystic fibrosis are linked to defective elimination of *Aspergillus* conidia by ECs [46]. Furthermore, contact with germinating *Aspergillus* hyphae ECs triggers the p38/ERK1-2 pathway [47]. The airways, through mucociliary clearance, remove conidia, explaining why patients with cystic fibrosis and bronchiectasis colonization may develop colonization or infection. ECs and alveolar macrophages (AM) are the first line of defense against aspergillosis. They must phagocytize and kill conidia although hyphae can pass through epithelium without altering its integrity [48]. *Aspergillus* conidia trigger Dectin-1, which recognizes the *Aspergillus* FleA fucose-binding lectin [44,49], and Dectin-2 activation in dendritic cells and AM, increasing fungal clearance [50]. After phagocytosis, the elimination depends on the generation of NADPH-dependent reactive oxidant species (ROS) [51] because a defect in this pathway in patients leads to invasive *Aspergillus* infection [52]. Furthermore, other pathways may be involved in pathogenesis and disease progression. For instance, patients who have received stem-cell or solid-organ transplants and Bruton’s tyrosine kinase inhibitors are susceptible to *Aspergillus* infection [35].

Besides AM, neutrophils are active against *Aspergillus*. Neutrophils are the primary immune effector against *Aspergillus* because severe neutropenia is a risk factor for the development of invasive aspergillosis. Neutrophil recruitment depends on chemokine release from lung cells [53]. The early phase lung ECs induce CXC chemokine release via IL-1B/MyD88/IL-1R-dependent, and in the late phase, the neutrophils migrations are driven in response via a CARD-9-dependent CXC chemokine [54,55]. Murine neutrophils also phagocytose and kill *Aspergillus* conidia via Dectin-1 and CARD-9 depending on Syk-kinase signaling and downstream NADPH oxidase-induced ROS production [56]. In addition, neutrophils release antimicrobial peptides, proteases, and neutrophil extracellular traps (NET) to avoid fungal invasion [57]. The ROS also mediates release of neutrophil extracellular traps (NETs) [58]. In addition, the adaptive immune system, such as T cells, is essential to the host’s defense. Both CD4 and CD8 cells provide protective immunity [59,60]. A type 2 helper T-cell response is linked to allergic diseases, whereas a proinflammatory phenotype is related to chronic pulmonary aspergillosis [30].

Host microbiome fluctuations may alter the host immune response facilitating fungal infection during immunosuppression periods [61]. Moreover, *Aspergillus* can contribute to pulmonary hypoxia by inhibiting angiogenesis [62]. During infection, the continuous activation of the inflammatory response contributes to the further development of hypoxia due to the destruction of the pulmonary tissue as collateral damage [63]. Indeed, strain fitness in low oxygen correlates well with *Aspergillus fumigatus* virulence [64]. ROS is critical in the host’s defense against *A. fumigatus* [65]. Moreover, increased ROS production is associated with disease severity in cystic fibrosis patients [66]. *A. fumigatus* express enzyme catalases [67], superoxide dismutases [68], and glutathione [69], which counteract ROS produced by the host. In addition, GAG triggers apoptosis in neutrophils via an NK cell-dependent mechanism [70] and inhibits neutrophil chemotaxis [71]. *Aspergillus* cell-wall-associated GAG mediates resistance to NADPH oxidase and augmented resistance to NETs [71]. In the animal model of IPA, GAG inhibits the protective Th1 response inducing Th2, which facilitates fungal infections. In human PBMCs, GAG dampened Th1 and Th17 cytokine production [72].

*Cryptococcosis* is an invasive fungal infection caused mainly by *Cryptococcus neoformans* or *Cryptococcus gattii*, often linked to AIDS, but it affects immunocompromised and immunocompetent patients [73]. Inhalation of *Cryptococcus* into the respiratory system is the main route of acquiring human infection, yet many clinicians mostly ignore pulmonary cryptococcosis [74]. *C. gattii* causes pulmonary infection in immunocompetent hosts [75]. Cryptococcosis caused by *C. neoformans* with initial pulmonary infection spreads to the central nervous system, becoming life-threatening in immunocompromised hosts [76]. Patients with neoplasia, transplants, or under treatment with immunosuppressant drugs are the most affected [77].

*Cryptococcus* produces melanin and the polysaccharide capsule, which harms macrophage phagocytosis, depletes complement, and impairs leukocytosis [78]. They also produce invasins and antioxidant molecules essential for dissemination to the central nervous system from the lungs [79]. In addition, the urease neutralizes the acidic pH of the phagolysosome and enables evasion [80,81]. Protective cell-mediated immunity is based primarily on *Cryptococcus*-specific Th1-type CD4^+^ T cells, which produce IL-2, IFN-γ, and TNF-α. A Th1–Th17 cytokine profile is associated with increased phagocytic activity and inhibition of cryptococcal growth, while a Th2 response is nonprotective [82]. Cryptococci induce a decrease in MHC class II-dependent antigen presentation and shift the immune response to a Th2 response, which decreases cryptococcal elimination [83]. In addition, recent work identified CPL1 as an effector of *C. neoformans*, which activates TLR4 signaling to drive the phosphorylation of STAT3 in macrophages, which polarizes interstitial macrophages during infection [84].

*P. jiroveci* affects many immunocompromised hosts, including AIDS, hematopoietic malignancies, and stem cell recipients [85]. *P. jiroveci* pneumonia develops via airborne transmission or reactivation of inadequately treated infection. Pneumocystis pneumonia prevalence is high in patients receiving immunosuppressants, notably with corticosteroids [86]. The concomitant COVID-19 and *P. jirovecii* pneumonia infection in immunosuppressed individuals or long-term corticosteroid therapy are associated with poor outcomes [87]. C-type lectin receptors (CLR) and TLR have been involved in AM inflammatory mediators’ production and the killing of *Pneumocystis*. AM binds *Pneumocystis* via a mannose receptor [88]. In addition, M1 macrophage polarization is favored in immunosuppressed mice, further reducing organism elimination [89]. Pneumocystis induces Th1, Th2, Th17, and T-cell regulatory responses. However, the precise role of each remains to be unveiled. The CD4^+^ T cells are critical to fighting *Pneumocystis* because losing them increases susceptibility to lung infection [90,91].

All discussed fungi possess immune system evasion mechanisms and induce similar host immune responses. In addition, the colonization or disease development will be dependent on the fungal furtive mechanism and host immune response. Further studies on host immune response may clarify the roles of cell surface receptors and the mechanism by which immune cells destroy pathogens.

## 3. Current Treatment for Fungal Pulmonary Diseases

Patients with acute or chronic fungal infections need the correct antifungal treatment for a successful outcome [92]. However, the available treatments for fungal infections are threatened by the few classes of antifungal drugs and the increasing resistance to a single drug or even multiple drugs by species such as *Candida* and *Aspergillus* [92]. Besides that, when related to pulmonary infections, there are some challenges in penetrating the lung tissue. Hydrophilic antimicrobials have tissue distribution limited to extracellular space, and they are found in extravascular lung water, but for relevant lung tissue intracellular penetration, the lipophilic drugs are most important [93]. Additionally, bioavailability should be maintained at least in the minimum inhibitory concentration. However, the lungs have adequate clearance by macrophages and mucociliary clearance [94]. Nonetheless, lungs benefit from pulmonary delivery via inhalation, where high local concentrations of active pharmaceutical ingredients can be delivered with no first-pass effect and a lower burden for the rest of the body [95]. Still, classical antifungals such as amphotericin B (AmB) and triazoles are not good candidates for nebulization and should preferably be delivered intravenously due to their physiochemical properties and pharmacokinetic characteristics [96].

Three primary antifungal agents are currently available to treat pulmonary fungal infections: triazoles, polyenes, and echinocandins [97]. Understanding the differences between dimorphic endemic fungi, filamentous fungi, and molds with respect to antifungal susceptibility and prognosis makes a definitive microbiologic or pathologic diagnosis imperative before treatment [98]. Regarding fungal pulmonary infection, treatment can range from non- to surgical resection, as in simple aspergilloma, the cause of CPA [99]. The fact that there are only a few classes of drugs and modes of action strictly limits treatment before the emergence of drug resistance. Species such as *C. auris* and *C*. *glabrata*, which are resistant to all antifungal classes, are increasingly reported [100]. Multidrug resistance challenges the treatment and outcome of the patient since it renders the antifungal arsenal useless. The intrinsic or acquired resistance mechanisms are target site modification and/or overexpression, target abundance, increased drug efflux, biofilm formation, and non-target effects [101].

An antifungal stewardship plan that considers local guidelines and diagnostic tests to determine when therapy should start and stop is crucial to preserve the future effectiveness of antimicrobials and improve patient outcomes. Therefore, a team of disease specialists, microbiologists, and infectious diseases pharmacists must have experience in fungal epidemiology and susceptibility patterns and the spectrum and pharmacokinetics of antifungal drugs and antifungal toxicities because it is essential to implement prescribing restrictions when these are no longer necessary [92,102]. Furthermore, the presence of programs that aim at better diagnosis and treatment is essential to avoid increasing fungal species resistance.

### 3.1. Azoles

The antifungal azole drug class is named according to the number of nitrogen atoms in the azole ring and is composed of imidazoles (clotrimazole, ketoconazole, miconazole), the first developed azoles. However, due to their high systemic toxicity, severe side effects, and interactions with other drugs, they were replaced by triazoles (fluconazole, itraconazole, voriconazole, posaconazole, isavuconazole) [103]. Azole compounds target the cytochrome P450 enzyme sterol 14α-demethylase, which converts lanosterol to ergosterol (Figure 1a). Inhibition of 14α-demethylase is fungistatic in yeasts and fungicidal in molds. Ergosterol is encoded by *ERG11* in yeast and *Cyp51* in molds [92] and is a fungal-specific sterol component of fungal cell membranes that determines the fluidity, permeability, and activity of membrane-associated proteins [104]. In treating pulmonary fungal infection, the important azoles are the triazoles, predominantly Itraconazole, and Fluconazole (Table 1).

Itraconazole was the first orally bioavailable drug with anti-*Aspergillus* activity [115]. *Aspergillus* is responsible for three classes of pulmonary aspergillosis, namely ABPA, CPA, and IPA [116], besides *Aspergillus bronchitis* [117]. Itraconazole is the first treatment against chronic cavitary pulmonary aspergillosis and ABPA and severe asthma with fungal sensitization [116]. It is also used against acute pulmonary and chronic cavitary histoplasmosis (*Histoplasma capsulatum*) [108], chronic pulmonary coccidioidomycosis caused by *Coccidioides*, as a secondary treatment [107], Paracoccidioidomycosis caused by *Paracoccidioides* [109], and Blastomycosis caused by *Blastomyces* [106]. The recommended daily dose of itraconazole is 200 to 400 mg. Therapeutic drug monitoring is recommended due to side effects such as congestive heart failure, cardiac effects, and drug interactions [118].

Fluconazole has a narrower-spectrum activity than Itraconazole, but it is used as primary treatment for mild-to-moderate pneumonic Coccidioidomycosis disease (800 mg/day) [107] and Cryptococcal pneumonia (400 mg/day) [111]. Moreover, it is active against many medically necessary *Candida* spp. except *Candida krusei* and has reduced or absent activity against *Candida glabrata* [119]. In addition, there are other Triazoles used as a second alternative to treatment or as prophylaxis for invasive fungal diseases. For example, Posaconazole is used as an alternative therapy for chronic pulmonary aspergillosis, histoplasmosis, and coccidioidomycosis, while voriconazole is used as an alternative for cryptococcal pneumonia and *Aspergillus bronchitis* [120].

### 3.2. Polyenes

The polyene amphotericin B (AmB) deoxycholate, introduced in 1958, was the first antifungal agent available to treat invasive fungal diseases [121]. Due to the intrinsic toxicity of AmB, it was necessary to develop novel, less-toxic lipid-based polyene formulations [118], and the liposomal amphotericin B composition is used in pulmonary fungal infections. Even though AmB has been widely used for more than 50 years, there are still findings about its mechanism. It was believed that AmB exerted cytocidal activity by forming ion channels inside lipid bilayers that permeabilize and kill cells [122]. However, more recent reports demonstrated that AmB exists primarily in large, extramembranous aggregates that exert fungicidal effects by extracting ergosterol from lipid bilayers [123]. More recent findings suggested that AmB extracts ergosterol but not cholesterol from the bilayers, causing membrane thinning [124]. Nowadays, it is known that the atomistic interactions that underlie fungicidal sterol sponges formed by Amb are asymmetric homodimers with large void volumes analogous to the size of sterols (Figure 1c) [125]. AmB has a broad-spectrum antifungal effect, covering most *Candida* spp. except *Candida auris*, *Candida haemulonii*, and *Candida lusitaniae* and *Cryptococcus* spp. and *Aspergillus* spp., with exception of *Aspergillus terreus*, *Histoplasma capsulatum*, *Coccidioides immitis* and *posadasii*, and *Paracoccidioides* spp. [121,126]. Despite the broad spectrum regarding pulmonary fungal infection treatments, AmB is most used as an alternative second-line treatment. Against coccidioidomycosis, it is used as AmB as rescue therapy, and during pregnancy [127], in chronic pulmonary aspergillosis, it is used as fourth-line treatment [105] and second-line for paracoccidioidomycosis severe cases [128]. It is the first choice for (i) acute pulmonary histoplasmosis [108]; (ii) chronic disseminated disease with diffuse alveolar lesions and cavitary disease caused by *Penicillium marneffei* [114]; and (iii) severe pulmonary sporotrichosis caused by *Sporothrix schenckii* [110].

### 3.3. Echinocandin

When fungal cell wall components and synthesis were discovered, echinocandins emerged, which target cell wall rather than cell membrane components. Echinocandins are an exceptionally safe treatment because their molecular target is not shared with mammalian cells [129]. Its mechanism inhibits β(1,3)-D-glucan synthase in a noncompetitive manner by forming a membrane-bound protein complex, specifically at the Fks1p extracellular subdomain (Figure 1b) [130]. Inhibition of β(1,3)-D-glucan synthase disrupts the structure of growing cell walls, resulting in osmotic instability and fungal cell death [103].

Caspofungin, micafungin, and anidulafungin are semisynthetic cyclic lipopeptides with a core composed of a cyclic hexapeptide with a lipophilic side chain [131]. This lipid residue is required to attach the drug to the cell membrane and is essential for bioactivity [132]. The Echinocandins spectrum covers *Candida*, *Aspergillus* species, and other pathogenic fungi [133]. Regarding pulmonary infections, echinocandins treat invasive aspergillosis but only when voriconazole is not tolerated or combined with voriconazole to enhance antifungal activity [105,129].

## 4. Amphotericin B Mechanism of Action and Challenges

AmB is a highly lipophilic antibiotic with low solubility and permeability [134]. In addition, it is unstable in pH ranges below 6 and above 9 and in the presence of light, heat, and oxygen, which makes it a challenging molecule considering any route of administration [135]. Thus, the first formulation, a micellar complex with sodium deoxycholate, allowed the stabilization of AmB and its use in patients in hospital environments through slow intravenous infusion [136,137]. However, this formulation has several adverse effects from fever, weight loss, nausea, and vomiting [138] to the most severe effect of prolonged use, which is nephrotoxicity, reported in 80% of patients [125], of which 50% evolve to acute renal failure [139,140]. A liposomal formulation of AmB was developed to solve this problem, which reduced the incidence of nephrotoxicity. However, new adverse effects emerged in addition to the high cost and the need for administration by slow intravenous infusion with hospitalization, which has been the cause of treatment refusal in approximately 76% of cases [136].

Despite this toxicity profile and the difficulties of use, AmB is one of the most potent drugs in treating visceral leishmaniasis and fungal diseases [137]. Furthermore, it is the first line in treating opportunistic and progressive fungal infections prevalent in immunocompromised patients, such as those with leukemia, the acquired immunodeficiency virus (HIV), and the intensive use of broad-spectrum antimicrobials or glucocorticoids [3,4,5,136,141].

AmB is used to treat invasive fungal diseases due to its broad spectrum and low incidence of resistance [9,142]. Its activity was first described in 1955, and the molecule was isolated from the bacterium *Streptomyces nodosus* [143]. It is a non-aromatic polyene antibiotic with amphoteric and amphiphilic character, forming soluble salts in alkaline and acidic media [9]. Its activity as an antifungal is in its interaction with fungal wall sterols, ergosterol, and the consequent disruption of the membrane [9,144], as shown in Figure 1. Actions within the cell are also described, such as the emergence of ROS, electrochemical imbalance with the opening of several ion channels, and disruption of mitochondria [9]. However, in the aggregated state, the AmB molecule loses its affinity for ergosterol and increases its affinity for cholesterol, causing severe toxicity, the most described being nephrotoxicity [9].

The first formulation, a micellar complex with sodium deoxycholate, allowed the stabilization of AmB and its use in patients in hospital environments through slow intravenous infusion [136,137]. Still, this formulation has several adverse effects, the most severe being nephrotoxicity [139,140]. The nanoformulations helped in this strategy, as they guarantee a release in the monomeric and controlled form, reducing the incidence of this adverse effect [9]. Moreover, liposomal AmB is the most used one, as it decreases the incidence of nephrotoxicity, and other side effects such as dyspnea, chest pain, feeling of death, and hypotension, among others, are described and are drawbacks in addition to the high cost and continuity of administration by slow intravenous infusion with hospital internment [145,146]. 

Due to the difficulty of achieving adequate concentrations of AmB in the lung by intravenous administration and seeking to minimize adverse effects, the formulations available on the market nebulized directly into the lung have been used [3,147]. The pulmonary pathway has advantages such as its rapid onset of action, low metabolic activity at the site of action, ability to avoid first-pass hepatic metabolism, lower risk of adverse effects, and thin thickness of alveolar epithelium [148].

## 5. Strategies for Pulmonary Delivery

Lung delivery is a complex process that depends on the formulation and patient characteristics. For pharmacological effect, drug particles must overcome the inferior airway, to be deposited deeper into the lung. This process depends on the particles’ size distribution, morphology, and density since these factors directly impact particle aerodynamics. The parameter that considers all these factors is the mass median aerodynamic diameter (MMAD). Ideally, particles must have MMAD equal to 1–2 µm to guarantee a deposition in the deep lung [149,150,151,152]. This information is critical when planning the characteristics of the final product. In the *Aspergillus fumigatus*, for example, the spores have sizes between 3 and 5 µm [153]. Hence, the particles in the delivery system should have similar sizes to reach the same regions in lung tissue. 

When the drug particle reaches the deep lung, the drug must dissolve in the pulmonary fluids to achieve the therapeutic effect. The solubilized drug can then reach the alveolar epithelium. There, it interacts with its target receptor, exerting its therapeutic influence. The effect’s amplitude depends on the drug’s retention in the alveoli [150,154]. Different devices can be used for pulmonary drug delivery. The most commonly used are nebulizers, pressurized metered-dose inhalers (pMDI), soft mist inhalers (SMI), and dry powder inhalers (DPI) [155]. Air-jet nebulizers are the leading inhaled device for delivering a large volume of aqueous conventional [156]. However, one strategy problem is that the drug output is lower than the total mass output. Thus, it is necessary to increase de concentration of the drug or a long period of nebulization [157,158,159]. In addition, viscosity, surface tension, and formulation composition directly impact the aerosol droplet size [160,161].

Tiddens et al. (2014) reviewed the topic of inhalational formulations for antibiotics, focusing on comparing liquid and dry systems. The authors argued that there is no way to determine an ideal formulation for all cases in the “one size fits all” style. Ideally, each challenge should be treated individually since the results will depend on several factors, such as disease, patient age, health status, and a more precise location for release. Moreover, even when deciding whether the administration will be in the form of dry powder or inhalation, it is worth noting that there are different devices for each, hence making development planning more complex. Therefore, following the concept of “target product profile”, for example, several parameters should be carefully considered from the beginning of a new project [162]. Some studies have indicated this in the case of AmB [163,164,165]. 

Due to the high complexity of the formulation for pulmonary delivery, regulatory entities such as the Food and Drug Administration (FDA), European Medicine Agency (EMA), and National Health Surveillance Agency from Brazil (ANVISA) have published guides to help the development of inhaled products [166,167,168]. However, nanocarrier systems are being widely studied in the literature because of the challenge of producing suitable formulations for pulmonary delivery [169,170]. Some advantages of these formulations are the ability to distribute the drug on the alveoli uniformly, enhance drug solubility, control release, increase targeting efficiency, and reduce the incidence of side effects [169,171,172,173,174]. In the pharmaceutical field, nanoparticles can range from 1 to 1000 nm [175,176,177]. It is worth pointing out that depending on the route of administration and the objective of the nanoformulation, this range can be more restricted or wider [169]. One drawback of using nanoparticles is that due to their small size, there is a higher surface interaction and a higher chance of particle aggregation, particle settlement, chemical instability, and difficulty dispersing the freeze-dried nanocarriers. Therefore, the use of nebulization to deliver colloidal dispersions is still a challenge [169,177]. Furthermore, the nanosystem does not deposit efficiently in the lungs due to its size, so most of it is exhaled. Therefore, the standard approach is to micronize powder carriers containing nanoparticles or agglomerated nanoparticles for delivery using MDIs and DPIs [177,178]. 

The most common nanocarriers used for pulmonary delivery are shown in Figure 2 [169,176,177]. Polymeric nanoparticles are usually produced by biodegradable polymers that can be synthetic (e.g., poly(ε-caprolactone), poly(lactic acid), poly(lactic-co-glycolic acid)) or natural (e.g., alginic acid, gelatin, chitosan) [179,180]. They usually form a spherical structure that can be massive (nanosphere) or have an oil or aqueous core (nanocapsule) [180]. Due to their biocompatibility, the capability of surface modification, drug protection, and sustained-release properties, polymeric nanoparticles are widely observed in the nanomedicine literature and can be applied to pulmonary delivery [176,177,181,182,183]. In addition, cationic polymers, such as chitosan and polyethyleneimine (PEI), are commonly used in pulmonary nanoformulation to enhance muco-adhesion and decrease particle aggregation [157,184,185]. In addition, these polymers can be used as surface modifiers to improve lung retention of different nanoparticle types [157,184,185]. However, it is essential to point out that careful in vitro and in vivo lung studies are required to establish the polymer and its degradation product compatibility since they can affect the surfactant properties in the alveoli and, consequently, affect breathing [169].

Liposomes are lipid particles formed by bilayer phospholipids that self-assemble, creating a spherical structure with an aqueous core (Figure 2). The first approved nanomedicine was a liposome, and nowadays, several approved liposomal drug products are commercially available [186,187,188]. The drug can be inside the liposome, in the aqueous core, or dispersed in the bilayer phospholipid [187]. Liposomes can be formed by one layer (unilamellar) or have a multilamellar (MLV) structure, which can entrap higher quantities of the drug. These MLVs can have sizes varying from 500 to 5000 nm, which is more suitable for pulmonary delivery [169,187]. The primary advantages of liposomes are (i) their capability to carry both hydrophilic and lipophilic drugs; (ii) their biocompatibility since the same components of cells membrane produce them, (iii) their enhancement of the effective delivery of the drug and can reduce side effects; and (iv) the potential to modify their size, charge, and surface by changing the lipid mixture. Liposomes are commonly used for pulmonary delivery since it is possible to add lipid lung surfactant to their structure, enhancing its compatibility [9,157]. In addition, there is described in the literature the attachment of mannose and a monoclonal antibody, which enhances the alveoli macrophage recognition and enhances lung retention [157,158,159].

The most common liposomal formulation is delivered as an aerosol by nebulizers [189]. To reach the deep lung, the liposomal aerosols must have size between 3–5 µm [190]. However, the stability of the aerosol liposome depends on the bilayer component, the pressure, the droplet size, and the stability of the components and the drug in the liquid state since during the air-jet nebulizing, vesicle fragmentation and drug loss occurs [156,191]. Therefore, liposomal dry powder formulations are being extensively studied and have shown promising results, with some tested in humans [176,177,192,193,194]. Inhalation devices can deliver this formulation. However, it is essential to point out the challenges of drying the liposomal formulation since the heat can oxidate the membrane phospholipid, and the impact during the process can destabilize the vesicle and release the drug. Moreover, dry powder formulations for pulmonary delivery have other challenges such as aerodynamic size, lung deposition, and physicochemical properties [195].

Other nanocarriers are less used than liposome and polymeric nanoparticles. Solid lipid nanoparticles (SLN) are spherical particles of solid lipids (e.g., Compritol^®^888 ATO–Gattefossé, Saint-Priest France-, Precirol^®^ ATO5– Gattefossé, Saint-Priest France-, and glyceryl monostearate) stabilized by surfactant. It appears as an alternative to polymeric nanoparticles since its components are well-tolerated by the body [196,197]. Other advantages are their capability for large-scale production, free organic solvent production, capacity to carry a high amount of lipophilic drug, and protection of the drug [196,197,198,199]. In addition, SLN has a controlled-release profile, particularly for lung delivery, but faster in vivo degradation when compared to polymeric nanoparticles with high tolerability [169,197]. Therefore, this system is suitable for pulmonary inhalation by both nebulizer and dry powder formulation, but more research is necessary [197,200,201].

Nano-emulsions are a mixture of lipids in the core containing the drug stabilized by a thin layer of surfactant. It can vary from 10 to 500 nm, has good kinetic stability, and has high lipophilic drug solubilization [202,203]. Polymeric micelles are obtained by amphiphilic di-block or tri-block copolymers dissolved in an aqueous medium above its critical micellar concentration that spontaneously forms a spherical core-shell structure [204]. Therefore, it can carry lipophilic drugs with a size range of 10–200 nm, and its preparation is simple [203]. 

Both nano-emulsion and polymeric micelles were not thoroughly studied in the pulmonary delivery field. This happens probably due to the difficulty stabilizing the system and not allowing it to aggregate during aerosol formation. Furthermore, the high amount of surfactant may destabilize the alveolar function by changing the lung surfactant [169]. Therefore, nano-emulsions and micelles are mainly used for gene delivery and vaccine application by nasal and pulmonary routes [156,184,203,205,206].

## 6. Use of AmB Formulation for Pulmonary Delivery

Commercially available AmB parenteral formulation is used to treat pulmonary fungal infections. However, it is difficult to reach the minimum inhibitory concentration of this drug in the lung, with high mortality [3]. Thus, clinical studies report using those parenteral formulations of deoxycholate and liposomal AmB via the pulmonary route through nebulization [9,194,207,208,209,210]. This route is chosen due to the fast onset of action and the lower risk of adverse effects [3,147,148,211]. Animal studies showed that aerosolized AmB effectively treated and prevented *Aspergillus* spp. and mucormycosis with higher concentration in the lung and lower systemic distribution [145,212,213,214]. Additionally, in animal studies, liposomal AmB showed higher pulmonary retention and no change in the composition of surfactants present in the lung [214,215]. Thus, it is suggested that nanotechnological formulations could be safer for the pulmonary delivery of AmB.

In humans, inhaled AmB was first described in 1959 [216]. Regardless of the parenteral formulation adapted to be nebulized, it was observed that in doses between 5 to 15 mg twice a day, a sufficient amount of AmB reached the lung for antifungal treatment, and plasma concentration was undetectable [207,208]. Therefore, the risk of adverse effects on delivery of AmB is expected to be lower than from parenteral use. Furthermore, AmB deoxycholate was nebulized prophylactically in patients who underwent lung, bone marrow, and other neutropenic transplants, and a reduction was observed in the chances of infection by *Aspergillus* spp. [209,217,218]. Reports of the use of AmB by this pathway for treating patients diagnosed with aspergillosis, tracheobronchitis, and bronchopulmonary candidiasis are also described [2,8]. In addition, the AmB liposomal form was used for those treatments and other less common pulmonary fungal diseases, such as mucormycosis [219,220]. There are approximately 30 studies on the database Clinicaltrials.com for administering AmB via inhalation, most of which use deoxycholate or liposomal AmB. Only two of these studies use formulations prepared for pulmonary or nasal administration: one of inhalable liposomal powder and the other via the nasal route [9,194]. 

Although several studies have demonstrated the success of using inhaled parenteral formulations of AmB, the use of this drug by this route is still open, and there is no consensus in the medical field. Thereby, such use is recommended by the Infectious Disease Society of America to prevent pulmonary fungi infections in patients with neutropenia but not by the Infectious Disease Pharmaceutical Society [99,221]. The contradiction is that the formulations used are not specific for pulmonary use. Until now, randomized controlled studies have been inconclusive since different formulations, nebulizers, and doses were used. Therefore, there is no reproduction in aerosol particle size, and many of them have inadequate size to reach the deep region of the bronchial tree [99,221]. Moreover, because they are parenteral formulations of AmB, practical issues such as foaming, photosensitivity, and oxidation of AmB and formulation as well as deposition of material in the nebulizer are critical points in this approach [8,212]. Finally, the solutions used do not have the correct osmolarity or pH, so variations in the results are found [222]. Some studies identified adverse effects such as cough, bronchospasm, dyspnea, and nausea in patients treated with deoxycholate AmB administered by nebulization [223,224]. Part of these effects was linked to the toxicity of sodium deoxycholate in lung surfactants. However, some of these adverse effects were also observed in liposomal formulations [8,225].

Another problem reported in the studies is treatment adherence since administration by nebulizers requires a long time of use so that only 10–20% of the dose reaches the lung. Thus, the solutions that will be sprayed must contain a few milligrams (5–30 mg) for an adequate amount of AmB to reach the lung. Because of that, the treatment also becomes more expensive [8]. Therefore, in a letter to the International Journal of Antimicrobial Agents, Yu and colleagues (2017) highlighted the importance of developing inhalable AmB powders to treat fungal lung diseases [222].

The critical point is that liposomal AmB showed better pulmonary retention and no change in the composition of surfactants present in the lung. Therefore, many studies, including those with AmB, opt for a liposome approach since it is possible to produce the structure of this particle with components present in lung surfactants [9,203,226]. This way, it is suggested that nanotechnological formulations will be safer for delivering AmB via the pulmonary route.

## 7. Technological Alternative to Delivery of AmB to the Lungs

Table 2 shows some works using carrier systems to deliver AmB to the lung. All of them are based on nanotechnology approaches. This is probably due to the toxicity and low solubility of free AmB. For example, Burgess et al. [227] observed an increasing trans-epithelial K^+^ current through the Calu-3 cell when treated with a formulation of AmB in deoxycholate. However, pure deoxycholate did not cause any change. Therefore, they attribute the observed toxicity in Calu-3 cells to AmB, probably to its capability to create pores in the membrane of Calu-3 cells and enhance electroconductivity.

In addition, almost all the works that presented the aerosolized particle size reported sizes between 2–5 µm, which is the ideal value for reaching the deeper lung [227] and, as mentioned above, is the same size as the *Aspergillus* spores [3,10] (Table 2). Only the work of Vyas et al. (2009) presented particle droplet sizes smaller than 2 µm [158]. However, they measured the size after collecting the particles from the stages of the imprinting apparatus as the others calculated the size at the moment of the aerosolization, which is more realistic. All the works with AmB nanoparticles aerosolized in vivo presented higher lung retention and lowered serum concentration (Table 2). This shows the drug’s retention in the pathology location and the nanotechnology potential.

Nine works used a type of lipid nanoparticle (Table 2) because it is known that using AmB on these types of nanoparticles, such as liposomes, reduces AmB toxicity when applied intravenously [228]. Four works used liposomes to enhance AmB retention in the lungs, which is in agreement with the literature where a parenteral AmB formulation, Ambisome^®^, was shown to be safer than Fugizone^®^ [8,225]. The liposomes proposed are multilamellar vesicles with layers that enhance the AmB encapsulation [159]. Only one work conducted an in vitro release study [201]. DPI formulation equivalent to 2 mg of AmB was dispersed in 5 mL of distilled water and sonicated, then placed in the dialysis bag and immersed in 100 mL PBS (pH 7.4, 37 °C). They observed a release of 61.22 ± 5.70% of the AmB in 48 h, indicating a slow and controlled-release behavior. Drug-release data were best fitted with the first-order kinetic model. However, it is essential to consider that this study did not consider the airway’s particularities, such as the presence of specific surfactants and their volume. Therefore, a more careful approach needs to be taken.

Three of these works proposed a surface modification to enhance lung retention (Table 2). One used chitosan, a cationic polysaccharide that enhances the mucoadhesive and prevents particle aggregation during aerosolization because of the positive surface charge [157,229]. Therefore, the aerosolized particle stays within the expected size to penetrate deeper into the lung. In addition, chitosan enhances the AmB entrapment in this organ [157], which agrees with the literature [185]. In this same work, Albasarah et al. (2010) modified the production of liposomes by adding polymer during its preparation, forming proliposomes. This strategy enhances AmB entrapment and the respirable fraction. One possible explanation is that it changes the physicochemical properties of the fluid and impacts the output of the air jet nebulizers [157].

**Table 2 pharmaceutics-14-02707-t002:** AmB formulation for pulmonary delivery as described in the literature.

Formulation	Methodology	Lung Application	Drug Concentration	Average Size (µm)	Zeta Potential (mV)	VMD (µm)	MMAD (µm)	GSD (µm)	FPF (%)	RF (%)	In Vitro Release	In Vivo *Test*	Reference
Aerosolized, non-ionic surfactant vesicle with cyclodextrin	Lipid film and then freeze-drying	Air-jet nebulization	0.93–1.19 mg/mL	1.677 ± 0.310	−70.8 ± 2.9	NA	NA	NA	NA	NA	No	Yes	[230]
Nanodisk composed of PL and Apo A-I	Lipid-film	NA	0.6 mg/mL	0.008–0.01	NA	NA	NA	NA	NA	NA	No	No	[231,232]
Nano-emulsion Intralipid^®^	Sonication and vorttexing	Air-jet nebulization	87.46 ± 2.21% (21.86 mg)	~0.375	−22	5.00 ± 0.07	NA	NA	57%	88	No	No	[156]
Nano-emulsion Clinoleic^®^			80.7 ± 0.70% (20.19 mg)	~0.325	−34	4.41 ± 0.19			80%	90			
Proliposomal microparticles/nanoparticles, aerosolized	Co-spray drying of the drug and the phospholipids	DPI/Handihaler (Boehringer Ingelheim)	0.202 ± 0.118 mg/mg	1.105 ± 0.461	NA	NA	12.1 ± 5.2	3.5 ± 0.5	13.0 ± 1.6	32.4 ± 9.6	No	No	[226]
			0.146 ± 0.0.019 mg/mg	1.110 ± 0.317	NA	NA	5.3 ± 1.0	2.7 ± 0.2	22.4 ± 4.1	46.0 ± 4.3			
			0.155 ± 0.060 mg/mg	1.311 ± 0.581	NA	NA	2.2 ± 0.1	1.8 ± 0.1	46.8 ± 5.4	93.6 ± 0.7			
Solid lipid nanoparticles	Solvent emulsification-evaporation to obtain the SLN and then lyophilization with a solution of lactose	NA	NA	0.187 ± 0.120 (SLN)	−30.16 ± 1.60	NA	NA	NA	35.71 ± 1.81 (1% lactose) 53.96 ± 3.67 (5% lactose) 72.57 ± 4.33 (10% lactose) 54.99 ± 3.04 (15% lactose) 22.03 ± 2.53 (20% lactose)	NA	Yes	No	[201]
Liposome (MLV)	Lipid film rehydration followed by sonication followed by the coating process	Ultrasonic jet nebulizer	74.29 ± 2.3%	0.413 ± 0.046	−18.5 ± 3.8	0.285 ± 0.025 *	NA	NA	NA	NA	No	No	[158]
Emulsomes			79.79 ± 3.1%	0.384 ± 0.040	−14.3 ± 2.1	0.286 ± 0.021 *							
Liposome coated OPM			72.19 ± 2.1 %	0.487 ± 0.039	−26.3 ± 2.3	0.378 ± 0.033 *							
Liposome coated with EBA-2 mAb			77.39 ± 2.7%	0.435 ± 0.038	−20.2 ± 1.9	0.280 ± 0.025 *							
Emulsomes coated with OPM			71.89 ± 2.1%	0.455 ± 0.042	−24.2 ± 2.5	0.343 ± 0.030 *							
Emulsomes coated with EBA-2 mAb			77.29 ± 1.9 %	0.392 ± 0.040	−21.7 ± 2.5	0.318 ± 0.028 *							
MLV	Lipid film and rehydration followed by recovering	Pressurized packed systems based on chlorofluorocarbon aerosol propellants	78.2 ± 1.3%	2.56 ± 0.32	NA	2.27 ± 0.25	NA	NA	NA	NA	No	Yes	[159]
MLVs coated with OPM			77.3 ± 2.5%	3.15 ± 0.59		2.87 ± 0.47							
MLV coated with OPP			76.8 ± 3.4%	3.23 ± 0.62		2.96 ± 0.58							
MLV mixture with lactose	Modified reverse phase evaporation								16.8 ± 2.2	NA	No	No	[192]
Polymeric micelles	Solvent evaporation	Air-jet nebulization	1780 ug	0.150 ± 0.006	47.2 ± 6.5		-	-	41.2 ± 1.7	NA	No	No	[205]
MLV	Thin-film hydration followed by sonication	Air-jet nebulization	0.067 mg/mL	0.181 ± 0.0034	−8.36 ± 2.8	3.43 ± 0.87	95.6 ± 1.2	NA	56 (188.6 ± 8 µg AmB)	NA	No	No	[157]
MLV coated with 0.2% of chitosan			0.079 mg/mL	0.196 ± 0.0016	+11.5 ± 3.1	3.82 ± 1.66	93.8 ± 0.9		58 (229.7± 9 µg AmB)				
Proliposomes liposome	Ethanol-based proliposomes followed by sonication		0.047 mg/mL	0.172 ± 0.0072	−11.4 ± 5.9	2.46 ± 0.92	94.3 ± 2.1		53.1 ± 2.6 (177.1 ± 6 µg)				
Proliposomes liposome coated with 0.3% of chitosan			0.056 mg/mL	0.211 ± 0.0033	+22.9 ± 2.4	3.28 ± 0.87	93.1 ± 2.6		61.3 ± 1.9 (243.7 ± 11 µg)				

* Dynamic light scattering was used to obtain these measurements. APO-1, apolipoprotein A-I; mAb, monoclonal antibody; MLV, multilamellar vesicle; NA, information not available; OPM, O-palmitoyl mannan; OPP, O-palmitoyl pullulan; PL, phospholipid.

Another strategy was to coat with alveolar macrophage-specific mannose. These would anchor the liposome in this type of macrophage and enhance its effectivity against aspergillosis since the alveolar macrophages are considered the densest site for *Aspergillus* infections where the fungi reside grow [159]. Vyas et al. 2005 used two ligands: O-palmitoyl mannan (OPM) and O-palmitoyl pullulan (OPP). They present a lower airway penetration in vitro than in a drug solution, probably due to the liposome’s interaction with the upper airway [159]. Furthermore, a decrease in the size and drug entrapment in liposomal aerosols was observed, suggesting spontaneous liposome reformation during the aerosolization. However, both surface-modified liposomes enhance the AmB amount in the lung when analyzing its biodistribution in albino mice even 24 h post administration. Additionally, they presented lower concentrations in the liver, spleen, and serum compared to non-coated liposome and AmB solution. This likely occurs due to a higher uptake by the macrophages because of mannose receptors on the AM surface and to a lower reorganization by the Type II alveolar epithelial cells, which might cause the destabilization of the uncoated liposome [158,159].

The same research group proposed in another work to coat emulsomes with OPM [158]. Emulsomes are lipid particles composed of a solid lipid core surrounded by phospholipid bilayers between liposomes and oil-in-water droplets [233]. They observed a higher survival rate of the immunosuppressed rats infected with *A. fumigatus* when compared to emulsomes without the OPM on the surface [158]. Vyas et al. (2009) proposed the functionalization of emulsomes and liposomes with EBA-2 monoclonal antibody (mAb) in this same work. They observed 100% survival of the immunosuppressed rats with pulmonary aspergillosis. Thus, it shows better histopathology of the lungs and higher retention in the lung for the nanoparticle functionalized with the mAb compared to non-functionalized nanoparticles and those functionalized with OPM. This could occur due to specific mAb at the surface of nanoparticles that may result in site-specific drug localization to the fungus surface [158].

Alsaadi et al. (2012) also tested their nanoparticle in immunosuppressed rats with pulmonary aspergillosis. They combined two technological approaches, cyclodextrin and non-ionic surfactant vesicle (NIV), to form their nanosystem. Their results on the in vitro tests showed that a low concentration of AmB reached the deeper lung. However, in the in vivo test, it was possible to observe that retention of the NIV in the lungs was higher, even 120 min after administration, when compared to the non-encapsulated drug. Furthermore, in the treatment of rats infected with *Aspergillus fumigatus*, the inhalation of NIV encapsulating AmB was more efficient in decreasing the fungus level and colony formation in lungs when compared to Fungizone^®^ (intraperitoneal route) and Posaconazole (oral route)) [230]. All three studies that tested the nanoformulations in vivo used nebulized systems, and the dose used was around 1 mg/kg (Table 2). Further, the two that evaluated efficacy used in vivo aspergillosis models [158,159,230]. Therefore, evaluating this system in other pulmonary fungi models is imperative. Besides the different types of nanoparticles used and surface functionalization, the works in the literature are lacking in exploring a new approach to delivering AmB into lungs.

Only one work used the dry-powder-inhaled strategy produced by the spray-drying technique (Table 2). They observed that the higher the phospholipid amount, the higher the fine particle formed and the respirable fraction [226]. Further, the MMAD presented by the particles has a more suitable size to reach deeper lung structures [149,150,151,152]. For this pathway, Novartis Pharmaceuticals Corporation and Nektar Therapeutics produced an AmB inhalation powder (ABIP) called NKTR-024. Particle sizes were between 2 to 5 µm, which is ideal since it has similarity to spores of *Aspergillus fumigatus* and, therefore, deposits in the same spot in the lungs [234,235]. Kirkpatrick et al. (2012) tested the ABIP formulation in a guinea pig model of invasive pulmonary aspergillosis and proved its prophylactic efficacy. The animals improved in survival and reduced pulmonary fungal burden [234]. In a phase I clinical trial, it was observed that plasma AmB concentration remained below those typically associated with renal toxicity. AmB in epithelial lining fluid in the lung was 70.7 and 189 µg/mL for doses of 25 mg + 5 mg for maintaining the dose for four weeks and 50 mg + 10 mg for maintaining the dose for four weeks [194]. This high lung fluid concentration shows that a more significant fraction of the ABIP was delivered to the lung. In addition, the ABIP was considered well-tolerated by the patients [194]. Spirometry remained virtually unchanged, which differs from the results reported for other aerosolized AmB formulations (deoxycholate and amphotericin B/lipid complex) [194,236,237].

Generally, it is crucial to consider the restriction of more in-depth results in all published studies. The articles mainly focused on in vitro characterizations, specifically in aerosolization parameters. Even so, few paid any more detailed attention to such aspects, and most used one or two parameters, not adding more data to the results. In addition, of the contemplated studies, only one performed in vitro release tests; there was, however, no consideration of the biorelevance aspects of the trial. The in vivo tests were considered for only one work. In this case, there was no in vitro release, and therefore, it did not generate conditions for any attempt to seek a relationship between both conditions.

Liao and Lam (2021) emphasized that the studies leave significant gaps by focusing on the physicochemical aspects without considering the in vivo evaluation [238]. Our study corroborates this assessment, adding new results. Additionally, they considered that one of the most relevant aspects of developing this type of product concerns the pharmacokinetic aspects. In addition, the use of tools with an approach in terms of biorelevance and pharmacokinetic modeling, although increasingly recommended, including by regulatory bodies, has not been used in the case of studies using nanotechnology for pulmonary drug delivery. It should be noted that the AmB itself has been approached from this perspective, albeit for other routes of administration [239]. Another important in vivo characteristic is the presence of lung surfactant, which can be used to optimize pulmonary drug delivery and distribute poorly water-soluble drugs through the respiratory system. Hidalgo et al. reported the improvement of Tacrolimus release using a native purified porcine pulmonary surfactant [240]. It is possible because Dipalmitoylphosphatidylcholine (DPPC), the main phospholipid present in lung surfactant, can reduce the surface tension in a water solution and solubilize poorly water-soluble molecules [240,241]. Unfortunately, this strategy has not yet been described in the literature for AmB. Thus, it was not explored herein.

Besides those encouraging results, there is no product for pulmonary delivery of AmB registered with the FDA, and no clinical trial is under way in the ClinicalTrial.gov database. This does not come as a surprise considering the reduction in the innovative character of the pharmaceutical industry that has been seen in recent decades. The translational challenge is recognized and can be identified as one of the most significant barriers to be overcome. The pulmonary route, in turn, has proven to be relatively resistant to technological innovations, with most of the drugs currently available making use of traditional and already consolidated technologies. Drug delivery systems, particularly those based on nanotechnology, face difficulties in reaching the market for several reasons. However, there is also good news, such as the registration of Arikayce (based on amikacin to combat non-tuberculous mycobacterium), which uses nanometric liposomes. It is not yet, however, a dry powder. With this, a vast field of exploration is envisioned, but it will depend on different efforts so that new realities contribute to the therapeutic increase in the line of products discussed here.

## Figures and Tables

**Figure 1 pharmaceutics-14-02707-f001:**
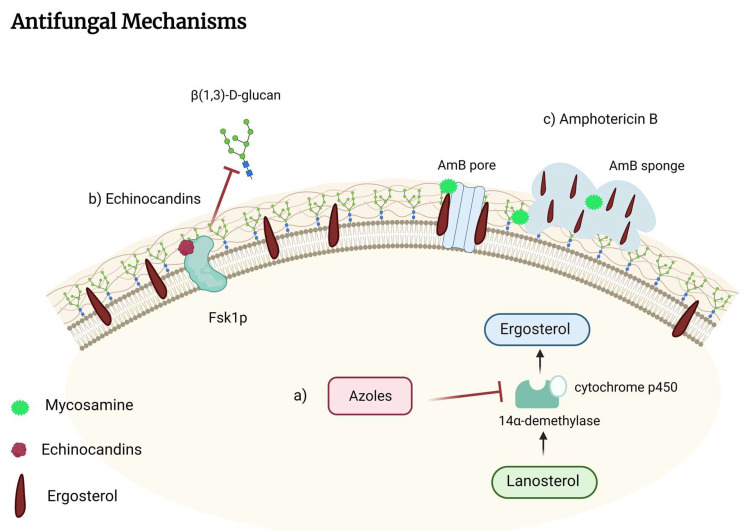
Mechanisms of action of antifungal drugs used in invasive pulmonary fungal infection. (**a**) Azole compounds target the cytochrome P450 enzyme sterol 14α-demethylase, which converts lanosterol to ergosterol. Inhibition of ergosterol biosynthesis leads to disrupted cell membrane integrity and fungistatic effect. (**b**) Echinocandins inhibits β(1,3)-D-glucan synthase, therefore inhibiting β(1,3)-D-glucan. β(1,3)-D-glucan constitutes more than 50% of the cell wall and is the main structural polysaccharide to which other cell wall components (chitins and glycoproteins) are attached. (**c**) AmB binds to ergosterol through mycosamine interaction, forming ion channels inside lipid bilayers. Another mechanism is the formation of sterol sponges that extract ergosterol from lipid bilayers.

**Figure 2 pharmaceutics-14-02707-f002:**
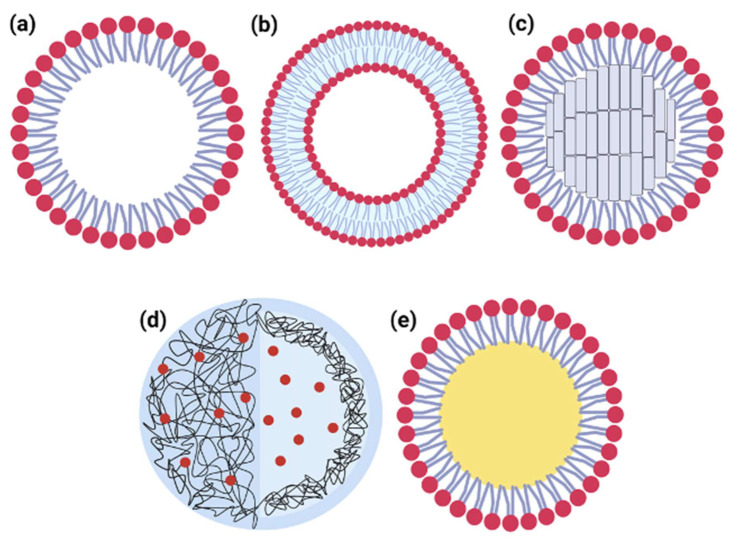
Types of nanoformulations. (**a**) Micelle, (**b**) liposome, (**c**) solid lipid nanoparticle, (**d**) polymeric nanoparticle, and (**e**) nano-emulsion.

**Table 1 pharmaceutics-14-02707-t001:** Therapies for pulmonary fungal disease.

Drug	Dosage Range	Delivery System	Mechanism	Fungal Lung Infection
Itraconazole	200–400 mg/day	Oral Intravenous	Inhibition of lanosterol 14α-demethylase	Aspergillosis [105] Fungal asthma (Allergic bronchopulmonary aspergillosis and severe asthma with fungal sensitize) Blastomycosis [106] Coccidioidomycosis [107] Histoplasmosis [108] Paracoccidioidomycosis [109] Sporotrichosis [110] Second-line: Cryptococcal pneumonia [111]
Fluconazole	50–800 mg/day	Oral Intravenous	Inhibition of lanosterol 14α-demethylase	Coccidioidomycosis [107] Cryptococcal pneumonia [111] Systemic candidiasis [112]
Voriconazole	200–400 mg/day 4–6 mg/kg/day	Oral Intravenous	Inhibition of lanosterol 14α-demethylase	Aspergillosis [105] Candidemia [112] Second-line: Cryptococcal pneumonia
Posaconazole	100–800 mg/day	Oral Intravenous	Inhibition of lanosterol 14α-demethylase	Second-line treatment: Invasive aspergillosis and coccidioidomycosis (refractory or intolerance to AmB or itraconazole or fluconazole). Prophylaxis of invasive Aspergillus and Candida infections [105,112]
Nebulized Amphotericin B	5–40 mg/day	Intravenous	Binding to ergosterol (Fungal membrane disruptor)	Fungal asthma (Allergic bronchopulmonary aspergillosis and severe asthma with fungal sensitization) [113]
Liposomal AmB	3–5 mg/kg/day	Intravenous	Binding to ergosterol (Fungal membrane disruptor)	Acute cavitary Histoplasmosis [108] *Penicillium marneffei* [114] Severe pulmonary sporotrichosis [110] Second-line: Chronic pulmonary aspergillosis [105] Chronic pulmonary Coccidioidomycosis [107] Paracoccidioidomycosis [109] Blastomycosis [106]
Echinocandins	50–70 mg/day	Intravenous	Inhibition of β(1,3)-D-glucan synthase	Aspergillosis (refractory or intolerance to AmB or itraconazole or voriconazole) [105] Candidiasis [112]

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
