# Peer review of "Could the Lung Be a Gateway for Amphotericin B to Attack the Army of Fungi?"

_pharmaceutics, 2022, doi:10.3390/pharmaceutics14122707_

Round 1
Reviewer 1 Report
Please see attached

Author Response
We would like to thank the opportunity to review our manuscript entitled “Could the lung be a gateway for amphotericin B to attack the army of fungi?” and reconsider it for publication in MDPI Pharmaceutics.
Also, we like to thank the reviewer for their comments. We believe that it certainly will contribute to improving the quality of the manuscript. We have answered all the reviewers’ comments and carefully evaluated the references to attend to the suggestions and improve the text quality. We hope we have addressed all raised issues.
Response to reviewer comments point-by-point:
Reviewer 1: In the manuscript entitled "Could the lung be a gateway for amphotericin B to attack the army of fungi?”. The authors have described the gateway for amphotericin B against fungal infections, especially aspergillosis. This manuscript is interesting and can be educational. By the way, there are some points such as old references that should be corrected.
Dear reviewer,
Thank you for the essential remarks aiming to improve the quality of our work. As seen below, in this letter, we answer each of the statements pointed out by the reviewers, and we hope they are enough to meet the requested requirements. The reviewers’ comments are colored black, while our replies are red.
Specific comments are mentioned below:
- Inside the text there are 228 references, but actually mentioned 126 references only!!
We are sorry that; it was an issue with our reference management software. We properly correct this issue.
- Lines 19-21: this sentence is unclear and need an English revision.
The authors would like to thank the reviewer for pointing out this issue and making the sentence clear. The grammatical and sentence formations were reviewed.
- Line 25: “antibiotic” to be antifungal.
We have appropriately corrected this.
- The abstract doesn’t have any conclusions!!
A conclusion was added in the abstract section, from the lines 31 to 35, as follows:
“Our finds indicate that despite an optimistic scenario for the pulmonary formulation of AmB based on the encouraging results discussed here, there is still no product registration on the FDA nor any clinical trial undergoing ClinicalTrial.gov.”
- Line 48, page 1: “…prolonged hospital stays, and AIDS.” this sentence doesn’t have any references. You can add new related articles.
We have adequately corrected it.
- Line 77, page 2: “Fungus Pulmonary diseases” to be Fungal Pulmonary diseases.
We have corrected this in Line 77, page 2 and line 2 page 4.
- Line 80, page 2: “…airborne transmitted and cause lung disease..” this sentence doesn’t have any references. Only for suggestion, you can add these new related articles: a. Assessment of indoor and outdoor airborne fungi in an Educational, Research and Treatment Center. Italian Journal of Medicine. 2017; 11: 52-56.
We have corrected it.
- Lines 82-84, page 1: “…will be dependent on the host's immune response.” this sentence doesn’t have any references. You can add new related articles.
We have corrected it.
- Lines 84, and 21 page 2: “Aspergillus fumigatus” to be A. fumigatus. And also, revised at rest inside all text for all organism names too.
The authors would like to thank the reviewer for pointing out this. Because “Aspergillus fumigatus” appears for the first time in line 21, page 2, we maintained the common name at this point. In the other parts of the text, “Aspergillus fumigatus” was replaced by A. fumigatus (including line 84, page 2).
- Line 93, page 2: “FleA” the abbreviations need an explanation!
We have corrected it.
- Line 28, page 3: “Cryptococcus neoformans” to be C. neoformans.
The authors would like to thank the reviewer for pointing out this. Because “Cryptococcus neoformans” appears for the first time in line 28, page 3, we maintained the common name in this point. In the other parts of the text “Cryptococcus neoformans” was replaced by C. neoformans.
- Line 45, page 3: “Pneumocystis jiroveci” to be P. jiroveci.
We properly correct this issue.
- Line 61, page 4: “…successful outcome.” this sentence doesn’t have any references. Only for the suggestion, you can add a new related article.
We have done that. The reference of this sentence is the same of the following sentence.
- Line 78, page 4: “Candida auris and glabrata” to be C. auris and C. glabrata. 2
We have done that.
- Lines 83-87: this sentence is unclear and needs an English revision.
The authors would like to thank the reviewer for point out this. The grammatical and sentence formations were reviewed as follows:
“The spores can propagate by transfer in air, colonizing the pulmonary airways and changing the status of the infection from asymptomatic to invasive. Clinical manifestations of disease will be dependent on the host's immune response [33,34]. The A. fumigatus complex followed by others as A. flavus may cause allergic bronchopulmonary aspergillosis (ABPA) or asthma with fungal sensitization (SAFS), whereas CPA affects susceptible people with lung diseases[32]. Immunosuppressed people may[31][35], which includes patients with cancer, autoimmune, or inflammatory diseases and present immune-metabolic abnormalities[36,37], also, in[38] Severe SARS-CoV-2 infection patients treated with immunosuppressants[39,40].”
- Line 45, page 6: “species” not to be Italic.
We have corrected it.
- Line 55, page 7: “Amphotericin B” to be AmB in the rest of the text.
We have corrected it.
- Line 11, page 8: “et al.” to be Italic in the rest of the text.
We have corrected it.
- Lines 5-8, page 10: this sentence is unclear and needs an English revision.
We have corrected it.
- Table 2, the abbreviations need an explanation! DLS!
We have corrected it.
- Lines 14-17, page 15: this sentence is unclear and needs an English revision. Não deixei marcado por que não encontrei – Juliana
We believe we have corrected it
- The end of the discussion needs complete conclusions according to valid results too!!
A conclusion was added in the discussion section, from the lines 808-815, as follows:
“This does not come as a surprise considering the reduction in the innovative character of the pharmaceutical industry that has been seen in recent decades. The translational challenge is recognized and can be identified as one of the most significant barriers to be overcome. The pulmonary route, in turn, has proven to be relatively resistant to technological innovations, with most of the drugs currently available making use of traditional and already consolidated technologies. Drug delivery systems, particularly those based on nanotechnology, face difficulties in reaching the market for several reasons. However, there is also good news, such as the registration of Arikayce (based on amikacin to combat non-tuberculous mycobacterium), which uses nanometric liposomes. It is not yet, however, a dry powder. With this, a vast field of exploration is envisioned, but it will depend on different efforts so that new realities contribute to the therapeutic increase in the line of products discussed here.”

Reviewer 2 Report
General comment
The manuscript “Could the lung be a gateway for amphotericin B to attack the army of fungi?” is well structured and interesting, since due to the increase in cases of immunocompromised patients associated with fungal infections, in recent years associated with the development of resistance to antifungals, treatment becomes highly relevant, particularly the use of amphotericin B, since It is a broad-spectrum antifungal, with the disadvantage of having adverse effects on the patient. Therefore, the authors discuss different strategies for releasing AmB in the lung through a potential nanoformulation of the drug in this work. However, I do have a few comments:
Line 43: I consider it appropriate to mention other fungi such as Histoplasma spp., Coccidioides spp., Paracoccidioides spp. Sporothrix spp. since antifungal therapy with amphoteric B has also been used in mycoses caused by these fungi, as shown in Table 1.
Line 77: I suggest mentioning the epidemiological aspects of Aspergillosis, Cryptococcosis, and Pneumocystosis to justify the choice of these mycoses and their causal agents, as the most important models that cause lung diseases.
Line 84: It would be convenient to clarify that pulmonary aspergillosis is mainly caused by Aspergillus fumigatus, however, it is also produced by other closely related species belonging to other sections.
References
Review the references and put them in the style of the journal, since in some the name of the journal is not abbreviated and there are upper and lower case letters in the titles of the articles.
Minor comments
Page 2 Line 77: Change “disease” to “Disease”
Page 7 Line 52: Change “Cell” to “cell”
Table2: Increase the size of the font corresponding to the references
Author Response
We would like to thank the opportunity to review our manuscript entitled “Could the lung be a gateway for amphotericin B to attack the army of fungi?” and reconsider it for publication in MDPI Pharmaceutics.
Also, we like to thank the reviewer for their comments. We believe it will undoubtedly improve the quality of the manuscript. We have answered all the reviewers’ comments and carefully evaluated the references to attend to the suggestions and improve the text quality. We hope we have addressed all raised issues.
Response to reviewer comments point-by-point:
Reviewer 2: The manuscript “Could the lung be a gateway for amphotericin B to attack the army of fungi?” is well structured and interesting since due to the increase in cases of immunocompromised patients associated with fungal infections, in recent years associated with the development of resistance to antifungals, treatment becomes highly relevant, particularly the use of amphotericin B, since It is a broad-spectrum antifungal, with the disadvantage of having adverse effects on the patient. Therefore, the authors discuss different strategies for releasing AmB in the lung through a potential nanoformulation of the drug in this work. However, I do have a few comments.
Dear reviewer,
Thank you for the significant remarks aiming to improve the quality of our work. As can be seen below, in this letter, we answer each of the comments pointed by the reviewers, and we hope they are enough to meet the requested requirements. The reviewers’ comments are colored in black, while our replies are in red.
- Line 43: I consider it appropriate to mention other fungi such as Histoplasma, Coccidioidesspp., Paracoccidioides spp. Sporothrix spp. since antifungal therapy with amphoteric B has also been used in mycoses caused by these fungi, as shown in Table 1.
We have corrected the sentence accordingly.
- Line 77: I suggest mentioning the epidemiological aspects of Aspergillosis, Cryptococcosis, and Pneumocystosis to justify the choice of these mycoses and their causal agents, as the most important models that cause lung diseases.
We have mentioned epidemiological aspects with the most recent global prevalence information we found.
- Line 84: It would be convenient to clarify that pulmonary aspergillosis is mainly caused by Aspergillus fumigatus, however, it is also produced by other closely related species belonging to other sections.
We have corrected it.
- Minor comments
- Page 2 Line 77: Change “disease” to “Disease”
- Page 7 Line 52: Change “Cell” to “cell”
- Table2:Increase the size of the font corresponding to the references
The authors would like to thank the reviewer for pointing out these mistakes. We properly corrected all the mentioned points, and the size of the font corresponding to the reference was increased.

Reviewer 3 Report
In their manuscript, Ferreirra de Carvalho et al. reviewed the state of the art about delivering amphotericin B through the lungs. In general terms, the manuscript is well written and structured. I enjoyed reading it. However, I have the following minor comments that may be addressed before supporting its publication.
· I found several typos throughout the manuscrips. Please, have a careful look. Some examples:
o Page 1 - Line 19: immunocompromised patients. such Aspergillus
o Page 1 - Line 23: developing -> development
o Page 1 - Line 24: Amphotericin B (AmB) has a broad-spectrum 24 antifungal effect low incidence of resistance.
o Page 2 - Line 66 -> SpreadS -> spread
· Page 7 - Line 59: it is unstable in pH 259 ranges below and above -> of what?
· Page 8 - Line 03: “The solubilized drug can then permeate the mucus until the alveolar epithelium”. This sentence is confusing and induce to think that there is mucus in the alveolar spaces. It is widely known that the alveolar lining fluid is mainly composed of lung surfactant with absence of mucus. Please, modify accordingly.
· As a potential delivery strategy, I recommend the authors to look after the literature about the potential use of lung surfactant as a drug delivery system and consider to include it in the manuscript. It may be an interesting alternative to solubilise and deliver amphotericin B. Some references to start with:
o Hidalgo, A., Garcia-Mouton, C., Autilio, C., Carravilla, P., Orellana, G., Islam, M.N., Bhattacharya, J., Bhattacharya, S., Cruz, A. and Pérez-Gil, J., 2021. Pulmonary surfactant and drug delivery: Vehiculization, release and targeting of surfactant/tacrolimus formulations. Journal of Controlled Release, 329, pp.205-222.
o Garcia-Mouton, C., Hidalgo, A., Cruz, A. and Pérez-Gil, J., 2019. The Lord of the Lungs: The essential role of pulmonary surfactant upon inhalation of nanoparticles. European Journal of Pharmaceutics and Biopharmaceutics, 144, pp.230-243.
o Baer, B., Veldhuizen, E.J., Molchanova, N., Jekhmane, S., Weingarth, M., Jenssen, H., Lin, J.S., Barron, A.E., Yamashita, C. and Veldhuizen, R., 2020. optimizing exogenous Surfactant as a pulmonary Delivery Vehicle for Chicken Cathelicidin-2. Scientific reports, 10(1), pp.1-11.
Author Response
We would like to thank the opportunity to review our manuscript entitled “Could the lung be a gateway for amphotericin B to attack the army of fungi?” and reconsider it for publication in MDPI Pharmaceutics.
Also, we like to thank the reviewer for their comments. We believe it will undoubtedly improve the quality of the manuscript. We have answered all the reviewers’ comments and carefully evaluated the references to attend to the suggestions and improve the text quality. We hope we have addressed all raised issues.
Response to reviewer comments point-by-point:
Reviewer 3: In their manuscript, Ferreira de Carvalho et al. reviewed the state of the art about delivering amphotericin B through the lungs. In general terms, the manuscript is well written and structured. I enjoyed reading it. However, I have the following minor comments that may be addressed before supporting its publication.
Dear reviewer,
Thank you for the essential remarks aiming to improve the quality of our work. As can be seen below, in this letter, we answer each of the comments pointed out by the reviewers, and we hope they are enough to meet the requested requirements. The reviewers’ comments are colored in black, while our replies are in red.
- I found several typos throughout the manuscript. Please, have a careful look. Some examples:
- Page 1 - Line 19: immunocompromised suchAspergillus
- Page 1 - Line 23: developing -> development
- Page 1 - Line 24: Amphotericin B (AmB) has a broad-spectrum 24 antifungal effect lowincidence of resistance.
- Page 2 - Line 66 -> SpreadS -> spread
- Page 7 - Line 59: it is unstable in pH 259 ranges below and above -> of what?
- Page 8 - Line 03: “The solubilized drug can then permeate the mucus until the alveolar epithelium”. This sentence is confusing and induce to think that there is mucus in the alveolar spaces. It is widely known that the alveolar lining fluid is mainly composed of lung surfactant with absence of mucus. Please, modify accordingly.
The authors would like to thank the reviewer for pointing out these mistakes. We have corrected all the points mentioned.
- As a potential delivery strategy, I recommend the authors look after the literature about the potential use of lung surfactant as a drug delivery system and consider to include it in the manuscript. It may be an interesting alternative to solubilise and deliver amphotericin B. Some references to start with:
- Hidalgo, A., Garcia-Mouton, C., Autilio, C., Carravilla, P., Orellana, G., Islam, M.N., Bhattacharya, J., Bhattacharya, S., Cruz, A. and Pérez-Gil, J., 2021. Pulmonary surfactant and drug delivery: Vehiculization, release and targeting of surfactant/tacrolimus formulations. Journal of Controlled Release, 329, pp.205-222.
- Garcia-Mouton, C., Hidalgo, A., Cruz, A. and Pérez-Gil, J., 2019. The Lord of the Lungs: The essential role of pulmonary surfactant upon inhalation of nanoparticles. European Journal of Pharmaceutics and Biopharmaceutics, 144, pp.230-243.
- Baer, B., Veldhuizen, E.J., Molchanova, N., Jekhmane, S., Weingarth, M., Jenssen, H., Lin, J.S., Barron, A.E., Yamashita, C. and Veldhuizen, R., 2020. optimizing exogenous Surfactant as a pulmonary Delivery Vehicle for Chicken Cathelicidin-2. Scientific reports, 10(1), pp.1-11.
The authors would like to thank the reviewer for pointing out this issue and providing the opportunity to improve our paper. To attend to the reviewer suggestion, we added a sentence about the lung surfactant in section 7 (Technological alternative to delivery AmB to the lungs), lines 799-805, as follows:
“Another important in vivo characteristic is the presence of lung surfactant, which can be used to optimize pulmonary drug delivery and distribute poorly water-soluble drugs through the respiratory system. Hidalgo et al. reported the improvement of Tacrolimus release using a native purified porcine pulmonary surfactant [241]. It is possible because Dipalmitoylphosphatidylcholine (DPPC), the main phospholipid present in lung surfactant, can reduce the surface tension in a water solution and solubilize poorly water-soluble molecules [241,242]. Unfortunately, this strategy was not described yet in the literature for AmB. Thus, it was not explored herein.”
